# Combined MITOchondrial-NUCLEAR (MITO-NUCLEAR) Analysis for Mitochondrial Diseases Diagnosis: Validation and Implementation of a One-Step NGS Method

**DOI:** 10.3390/genes14051087

**Published:** 2023-05-15

**Authors:** Ferdinando Barretta, Fabiana Uomo, Filomena Caldora, Rossella Mocerino, Daniela Adamo, Francesco Testa, Francesca Simonelli, Olga Scudiero, Nadia Tinto, Giulia Frisso, Cristina Mazzaccara

**Affiliations:** 1Department of Molecular Medicine and Medical Biotechnology, University of Naples Federico II, 80131 Naples, Italycristina.mazzaccara@unina.it (C.M.); 2CEINGE Advanced Biotechnologies Franco Salvatore, 80131 Naples, Italy; 3Department of Neuroscience, Reproductive Sciences and Dentistry, University of Naples Federico II, 80131 Naples, Italy; 4Eye Clinic, Multidisciplinary Department of Medical, Surgical and Dental Sciences, University of Campania Luigi Vanvitelli, 80138 Naples, Italy

**Keywords:** mitochondrial diseases, MITO-NUCLEAR, next-generation sequencing (NGS), mtDNA, nuclear genes, mitochondrial and nuclear disease panel, combined sequencing of mitochondrial and nuclear DNA

## Abstract

Background: Next-generation sequencing (NGS) technology is revolutionizing diagnostic screening for mitochondrial diseases (MDs). Moreover, an investigation by NGS still requires analyzing the mitochondrial genome and nuclear genes separately, with limitations in terms of time and costs. We describe the validation and implementation of a custom blended MITOchondrial-NUCLEAR (MITO-NUCLEAR) assay for the simultaneous identification of genetic variants both in whole mtDNA and in nuclear genes included in a clinic exome panel. Furthermore, the MITO-NUCLEAR assay, implemented in our diagnostic process, has allowed us to arrive at a molecular diagnosis in a young patient. Methods: Massive sequencing strategy was applied for the validation experiments, performed using multiple tissues (blood, buccal swab, fresh tissue, tissue from slide, and formalin-fixed paraffin-embedded tissue section) and two different blend-in ratios of the mitochondrial probes: nuclear probes; 1:900 and 1:300. Results: Data suggested that 1:300 was the optimal probe dilution, where 100% of the mtDNA was covered at least 3000×, the median coverage was >5000×, and 93.84% of nuclear regions were covered at least 100×. Conclusions: Our custom Agilent SureSelect MITO-NUCLEAR panel provides a potential “one-step” investigation that may be applied to both research and genetic diagnosis of MDs, allowing the simultaneous discovery of nuclear and mitochondrial mutations.

## 1. Introduction

Mitochondrial diseases (MDs) (1:5000–10,000) represent a wide group of human disorders, extending from isolated organ involvement to complex multisystem diseases [1,2,3,4].

Clinically, MDs show an extremely heterogeneous phenotype, depending on the involved tissue, the specific mtDNA mutation, and its heteroplasmic level. Furthermore, numerous nuclear gene mutations are also associated with MDs. Mitochondrial diseases affect preferentially tissue with high-energy demands and give rise to several degenerative disorders, involving mainly neuromuscular, ophthalmological, and gastroenterological features [1,2,5,6,7]. In addition, deafness, diabetes, cardiomyopathy (in particular dilated, dCMP; hypertrophic, hCMP; restrictive, rCMP), and cardiac conduction defects are also frequently observed [8,9,10,11,12,13].

MDs appear in childhood and adulthood and are characterized by defects of oxidative phosphorylation (OXPHOS) due to both mitochondrial DNA (mtDNA) and nuclear genome (nDNA) alterations impairing ATP production [14,15,16]. Moreover, estimates suggest that only about 15–20% of respiratory chain deficit (RCD) is due to mtDNA mutations, while the remaining part is caused by nuclear defects [9]. Particularly, more than a thousand proteins encoded by nuclear genes are responsible for the arrangement, expression, assembly, stability, and function of the five-multimeric OXPHOS subunits [1,2,17]. On the contrary, the mitochondrial genome encodes for 13 structural proteins of the OXPHOS system and 24 RNA (22 tRNA and two rRNA) [18]. Based on its function, the “Mitochondrial Proteome” can be approximately grouped into the following categories: OXPHOS subunits (both from the nuclear and mitochondrial genome), assembly factors, and electron carriers; mtDNA maintenance; mtDNA expression (including synthesis, processing, and modification of mt-rRNAs, mt-tRNAs, mt-mRNAs, mitoribosome biogenesis, and translation); enzyme cofactors; mitochondrial homeostasis and quality control (including mitochondrial protein import, lipid modification and homeostasis, mitochondrial morphology comprising fission/fusion and cristae organization factors, protein quality control, and apoptosis/autophagy) and general metabolism [18,19]. The twofold genetic control on the mitochondrial metabolism, explains the different pattern of MDs inheritance, including maternal, X-linked, autosomal recessive, autosomal dominant. *De novo* mutations are also described in [19].

The mtDNA is often initially sequenced in patients with a clinical diagnosis and a strongly suggestive family history of MDs caused by alterations of mtDNA to exclude a primary defect of mtDNA. However, due to the many nuclear genes involved in the MDs, which are impossible to investigate by the Sanger approach, many patients still lack a molecular diagnosis.

Currently, next-generation sequencing (NGS) technology is revolutionizing diagnostic screening for rare diseases, allowing the simultaneous analysis of many genes and samples showing good diagnostic sensitivity with low costs and optimized analysis times [20,21,22,23,24,25]. Next-generation sequencing technologies have improved the diagnosis of MDs, increasing diagnostic success from <20% to >60% [8]. Moreover, MD investigation by NGS typically still requires analyzing separately mtDNA and several nuclear genes causing mitochondrial disease, with limitations in terms of time and costs [22,26,27,28,29,30,31]. Few studies have recently developed an NGS-based method with combined sequencing of the whole mtDNA and several nuclear genes [20,27,32,33]. In particular, Falk M.J and colleagues reported the performance characteristics of a custom Agilent whole-exome capture (1:100 Mito-Plus Whole-Exome) designed to facilitate simultaneous analysis of the standard 50 Mb whole exome, with optimized coverage of the complete MitoCarta (www.broadinstitute.org/pubs/MitoCarta) nuclear genes set and the mtDNA genome. In this paper, the 1:100 molar ratios of mitochondrial probes provided a dual-genome coverage, with 10× coverage for over 97.5% of all nuclear regions and 1000× coverage for 99.8% of the mtDNA genome [27].

Following the paper of Falk M.J. et al., we here describe the validation and implementation of a custom blended MITOchondrial-NUCLEAR (MITO-NUCLEAR) assay by using NGS for the simultaneous identification of genetic variants both in whole mtDNA (SSel Custom Constitutional Panel 16.708 Kbp) and in nuclear genes included in a clinic exome panel (CCP17: Custom Constitutional Panel; Design size: 17 Mb; 5227 genes). Taking into account the different nuclear design sizes, compared to the 50 Mb design by Falk M.J., we decided to investigate two different blend-in ratios of the mitochondrial probes 1:900 and 1:300 (mitochondrial probes: nuclear probes).

A parallel massive sequencing strategy, based on NGS technology, was applied for the validation experiments, performed using samples from multiple tissues (blood, buccal swab, fresh tissue, tissue from slide, and formalin-fixed paraffin-embedded FFPE tissue section).

## 2. Materials and Methods

### 2.1. Samples for Validation Experiments

To optimize the interrogation of nuclear and mtDNA genes, a custom Agilent SureSelect MITO-NUCLEAR panel was formulated by blending mtDNA and CCP17 probes from two separate designs (ChrMito hg38, design size: 16.708 Kbp for MITOchondrial genome and NUCLEAR clinic exome CCP17, hg19: Custom Constitutional Panel; Design size: 17 Mb for nuclear clinic exome). MITO-NUCLEAR validation experiments were performed using eight samples from multiple tissues.

Four samples (S1, S2, S4, and S5) from patients with suspected mitochondrial disease were investigated as positive controls for mtDNA variants [13]. These patients were previously analyzed by our study group by Long-PCR of mtDNA and subsequent sequencing by the Sanger method, which identified several mtDNA variants (Appendix A). In particular, S1 and S2 (brother and sister) were patients with maternally inherited diabetes and deafness (MIDD) clinical diagnoses. They showed diabetes, bilateral sensorineural hearing loss, myopathy, retinopathy, cardiomyopathy, and cerebrovascular disorders. Maternal family history for these features was also reported. S4 and S5 were buccal swab specimens from patients affected by mitochondrial diabetes.

The S3 sample came to the attention of the CEINGE_Biotecnologie Avanzate Franco Salvatore (Naples, Italy), which is the Unique Regional Center for Newborn Screening in Campania and Regional Reference Center of the National Health Service for Clinical Molecular Biology, Laboratory Genetics, for suspected metabolic disease. Genetic analysis, by NUCLEAR clinic exome CCP17, including the routine nuclear genes panel for the molecular diagnosis of metabolic disorders, did not show pathogenic variants, but only variants classified as polymorphisms and variants of uncertain significance (VUS), according to the American College of Medical Genetics and Genomics (ACMG) [34]. Following hypotonia, growth retardation, and lactic acidosis, the suspicion of MD was supposed. Thus, the S3 sample was analyzed by MITO-NUCLEAR assay, acting as a positive control for nuclear genes, as well (Appendix A). Furthermore, skeletal muscle samples from fresh tissue (TS1), tissue fixed to a slide (TS2), and FFPE tissue (TS3) were used to test the method on different tissue processing. Informed consent was obtained from all participants according to the Helsinki Declaration [35], and the internal ethics committee approval (N. 77/21) was obtained.

### 2.2. DNA Purification

Genomic DNAs from both peripheral blood samples with EDTA-K2 (S1, S2, S3) and buccal cells (S4, S5) were extracted using the Nucleon BACC3 kit (GE Healthcare, Hatfield, UK) and Puregene Buccal Core Kit A (Qiagen Science, Germantown, MD, USA), respectively. Genomic DNAs from fresh tissue, tissue from the slide, and tissue from FFPE (TS1, TS2, and TS3) were purified by Kit Qiamp fast Tissue and Qiamp DNA FFPE tissue Kit (Qiagen Science, USA). Before proceeding to library preparation, the concentration and purity of the DNAs were assessed both by fluorimetric method, using Qubit dsDNA HS Assay Kit (Thermo Fisher Scientific, Waltham, MA, USA) and by a spectrophotometric method by Nanodrop instrument (Thermo Fisher Scientific). The DNAs integrity was evaluated by ScreenTape analysis on TapeStation (Agilent, Santa Clara, CA, USA).

### 2.3. Samples Preparation

Given the natural excess of mtDNA, with respect to the nuclear genome, in terms of the molar ratio of mtDNA to nuclear DNA, it was necessary to dilute the mitochondrial probes to ensure optimal coverage output of both genomes, as also reported in the literature [27,32]. Marni J. Falk et al. suggested that 1:100 molar ratios of mtDNA to nuclear probes provided an optimal coverage for both nuclear regions and mtDNA genome, where over 99.9% of the mtDNA genome was covered at least 100×, over 99.0% of the mtDNA genome was covered at least 1000×, the median coverage was 7918×, and the minimum depth of coverage for any mtDNA base was 41×. In their study, Marni J. Falk et al. employed nuclear probes designed to amplify the whole exome (WES, 50 Mb), which is about three times larger than the clinical exome CCP17 used in our validation experiments. For this reason, in our study, we evaluated two different molar dilution ratios, 1:300 and 1:900 (mitochondrial probes: nuclear probes), in the RUN 1 assay, and then we selected the 1:300 molar dilution ratios for subsequent experiments (RUN 2). Samples preparation and hybridization were performed in duplicate to confirm the repeatability of the method.

In RUN 1, three samples (S1, S2, and S3) were hybridized with a mix of both probes at two different dilutions, and one sample (S2) was also hybridized with only nuclear probes to verify the coverage usually obtained without the mtDNA and CCP17 blended.

In the second validation experiment (RUN 2), seven samples (S1, S2, S4, S5, TS1, TS2, and TS3) were hybridized with 1:300 diluted probes. The S1 and S2 samples were re-analyzed as a positive control to confirm the previous dilution; samples S4 and S5 were analyzed for the first time in this validation experiment, and TS1, TS2, and TS3 samples were investigated with the aim of validating the method for samples extracted from other tissues than blood and buccal cells.

### 2.4. Next-Generation Sequencing

We used an NGS-based method, combining sequencing of the whole mtDNA and nuclear genes included in a clinic exome panel. For this purpose, NGS libraries were prepared from 100 ng of each DNA sample, using mtDNA genome probes (SSel Custom Constitutional Panel 16,708 Kbp; Agilent Technologies, Santa Clara, CA, USA) and CCP17 probes (Custom Constitutional Panel; Design size: 17 Mb; Agilent Technologies). All samples capture libraries were prepared as described in the SureSelectXT-HS Target Enrichment System for Illumina Paired-End Multiplexed Sequencing Platforms protocol. Sequencing sample pools was carried out on an Illumina NextSeq 550 system (Illumina, San Diego, CA, USA) as 150 bp paired-end runs using v2.5 chemistry. The reads alignment (BWA), the call (GATK), and the annotation (Annovar) of the variants were executed, respectively, using the Alissa Align & Call and Interpret (Agilent Technologies); the subsequent classification of variants was performed following the guidelines of the American College of Medical Genetics and Genomics (ACMG) [34] and MITOMAP (www.mitomap.org, accessed on 13 December 2022) for nuclear and mitochondrial variants, respectively.

## 3. Results

### 3.1. Libraries Samples Suitability

The final quality of the custom-blended MITO-NUCLEAR libraries was evaluated using the TapeStation system (Agilent Technologies) that allows evaluation of both the library’s concentration and the size of the DNA fragments. Overall, an average concentration of 122 ng/μL was obtained for samples extracted from blood (S1, S2, and S3) and buccal swabs (S4 and S5), whereas samples from tissues (TS1, TS2, and TS3) showed an average concentration of 148 ng/μL. For all samples, the size of the DNA fragments was between 200 and 400 bp, as expected by protocol (Figure 1).

### 3.2. Experimental Evaluation of the Capture Efficiency of mtDNA (SSel Custom Constitutional Panel 16.708 Kbp) and Nuclear CCP17 (Custom Constitutional Panel; Design Size: 17 Mb) in Blended MITO-NUCLEAR Panel at Different Concentrations

Eleven samples were sequenced into two experimental runs at two different molar ratios of mitochondrial and CCP17 probes. Overall, 27 bioinformatics analyses were performed with respect to the coordinates of the human genome hg19 (GRCh37.p13), while the analysis of the whole mitochondrial genome (NC_012920) was performed versus the human genome coordinates hg38 (GRCh38.p13). Coverage performances for the blended MITO-NUCLEAR, with varying molar ratios (1:300 and 1:900) of custom probes, are shown in Table 1.

Data analysis suggested that 1:300 was the optimal mtDNA:nuclear molar ratio, where 100% of the mtDNA genome was covered at least 3000×, and the median coverage was 5513×. The 1:900 molar ratios provided that 100% of the mtDNA genome covered at least 1000×, but only 80.6% of the mtDNA genome covered at least 2000×, and 63.8% covered at least 3000× (Figure 2).

Furthermore, whereas higher depth of coverage is required for the entire mtDNA genome analysis to allow a reliable detection of low heteroplasmy, lower median coverage is generally acceptable for nuclear analysis. Specifically, in our study, 93.84% of nuclear regions were covered at least 100× by using 1:300 molar dilution ratios (mitochondrial probes: nuclear probes), with a similar percentage (93.51%) by using a 1:900 dilution (Figure 2).

### 3.3. Diagnostic Yield of MITO-NUCLEAR Assay

To assess the diagnostic yield of the combined nuclear and mitochondrial DNA capturing method, we compared the efficiency of our MITO-NUCLEAR design to identify mtDNA variants with respect to standard procedures, such as PCR and Sanger sequencing.

All mitochondrial variants, including benign polymorphisms, previously identified in patients S1, S2, S4, and S5, by Long-PCR of mtDNA and subsequent Sanger sequencing, were confirmed by MITO-NUCLEAR assay. In fact, the MITO_NUCLEAR assay found 51 different variants previously identified by Sanger, and, of note, also two variants were missed by the traditional approach. The detail of the identified variants is shown in Appendix A.

Furthermore, for patients S1, S2, S4, and S5, no known mutations, but only benign variants and variants of uncertain significance (VUSs) were highlighted in the bioinformatics pipeline (Appendix A), investigating 257 nuclear genes from the CCP17 panel. These nuclear genes (Appendix A) were selected from those commonly reported in the literature and in MitoCarta, mtDB (http://www.mtdb.igp.uu.se/, accessed on 13 December 2022), and MITOMAP (https://www.mitomap.org/, accessed on 14 December 2022) databases as associated with mitochondrial disorders [3,26,31,32,36,37]. Finally, all nuclear variants previously detected in the S3 sample were confirmed by MITO-NUCLEAR analysis (Appendix A). Variant nomenclature is based on HGVS (http://www.hgvs.org/mutnomen/, accessed on 13 December 2022) and MITOMAP (https://www.mitomap.org/MITOMAP, accessed on 13 December 2022) guidelines.

### 3.4. mtDNA Genome Heteroplasmy Detection

The mtDNA genome, investigated by MITO-NUCLEAR NGS assay, showed two single nucleotide heteroplasmic variants: the m.16114 C > T in MT-HV1 region, with 71% of T base and 28% of wild type C base, Read Depth (RD): 4617, in S2 sample; and the m.3243 A > G *MT-TL1*, showing 41% of mutated G base and 57% of wild type A base, RD: 4305, detected in the S5 sample. Split/count data for both heteroplasmic variants are highlighted in Appendix A. Our previous study, performed on the S5 sample by real-time quantitative PCR (qRT-PCR), showed a similar heteroplasmy level (34%) [13]. These data confirmed that the determination of the heteroplasmy by NGS was consistent with the heteroplasmy determined by one of the most accredited quantitative methods.

### 3.5. Molecular Diagnosis of Autosomal Dominant Optic Atrophy by MITO-NUCLEAR Investigation (Case Report)

Following the MITO-NUCLEAR validation experiments, we applied our MITO-NUCLEAR assay in clinical diagnostic evaluation, investigating a six-year-old girl of Israeli origins who was referred with a history of bilateral slowly progressive vision loss to the Eye Clinic, Multidisciplinary Department of Medical, Surgical and Dental Sciences, University of Campania Luigi Vanvitelli, Naples, Italy. The ophthalmological assessment revealed a bilateral reduction of visual acuity of 20/200 in each eye. Anterior segment slit-lamp examination was unremarkable, and intraocular eye pressure was within normal limits. Fundus ophthalmoscopy revealed normal retinal reflexes and macular configuration, with bilateral optic disc pallor consistent with optic atrophy (Figure 3). Electroretinography was normal, and visual evoked potential (VEP) testing showed a bilateral increase in P2 latency (130 to 138 ms). Parents and grandparents were apparently unaffected.

Genomic DNAs from both peripheral blood samples with EDTA-K2 and buccal cells sample were obtained, as reported above. Next-generation sequencing was carried out as described in Section 2.4 by using the validated 1:300 molar dilution ratios.

The bioinformatics analysis of 257 nuclear genes from CCP17 showed an average read depth of 154×. A total of 98% of targeted nuclear regions were sequenced with a read depth greater than 25×. For the whole mitochondrial genome, an average read depth of 4713× was observed, and 100% of target regions were sequenced with a read depth greater than 1000×. Molecular analysis, by using MITO-NUCLEAR assay, highlighted a heterozygous known pathogenic mutation in the OPA1 nuclear gene (NM_130837.3) (c.2873_2876del; p.Val958GlyfsTer3; CD002708; rs80356530), while the mitochondrial genome investigation showed only known polymorphisms. The subsequent Sanger sequencing of the OPA1 gene mutation confirmed the OPA1 heterozygous mutation.

## 4. Discussion

Mitochondrial disorders are the most genetically heterogeneous group of diseases involving both nuclear and mitochondrial genomes. Therefore, the diagnosis of these disorders is still today a great challenge. The molecular diagnosis of mitochondrial DNA defects may utilize multiple methods for the detection and quantification of mtDNA point mutations and large deletions [38]. Among these methods, long-range PCR, followed by Sanger sequencing, are widely used [38,39,40]. However, this procedure requires several separate reaction steps that are challenging to integrate into the diagnostic workflow. This diagnostic approach could be followed only in cases with a clinical phenotype strongly suggestive of a specific mitochondrial pathology (e.g., Leber Hereditary Optic Neuropathy) for which one or a few mitochondrial genes are reported. Similarly, diagnosis of mitochondrial disorders caused by nuclear genes often requires the stepwise Sanger sequencing of the candidate genes one by one. Moreover, this approach may not identify the molecular alteration, leaving many MD patients without a diagnosis.

The development of next-generation sequencing (NGS) has revolutionized the diagnostic approach of MDs, allowing the massively parallel sequencing analysis of the entire mitochondrial genome, as well as the simultaneous analysis of a group of nuclear genes or the whole exome.

However, NGS-based mtDNA genome analysis blended with nuclear genes analysis is not widespread in diagnostic laboratories.

To date, only a few scientific studies have focused research on the possibility of combining sequencing of the complete mtDNA and nuclear genes in one “blended” NGS-based method. Moreover, whole genome sequencing approaches have recently been used to simultaneously sequence both mitochondrial and nuclear genomes [27,32,33,41].

Following pioneer studies of Falk M.J. et al. and Abicht A. et al., we evaluated the capture efficiency of the entire mtDNA and nuclear clinic exome CCP17, including 257 nuclear mitochondrial disease-related genes, reported in MitoCarta, mtDB, Mitomap databases, and the literature. We used a “blended” MITO-NUCLEAR panel at two different blend-in ratios of the mitochondrial probes, 1:900 and 1:300 (mitochondrial probes: nuclear probes), in the first validation experiments, and then we confirmed the 1:300 molar dilution ratios for the next ones. Several experiments were needed to evaluate the coverage performances for the blended MITO-NUCLEAR, with varying molar ratios (1:300 and 1:900); thus, the number of tested samples was not very large. Nevertheless, we, globally, confirmed about 600 variants (546 variants in nuclear genes from the S3 sample and 51 variants in mtDNA from S1, S2, S4, and S5 samples), providing good confidence in test sensitivity. The confirmation of all nuclear variants previously found in the S3 sample highlights that the mitochondrial probes blended with nuclear ones in MITO-NUCLEAR do not affect the performance of the assay. Moreover, we achieved an optimal dual-genome coverage, where 100% of the mtDNA genome was covered at least 3000×, and 93.84% of all nuclear regions were covered at least 100× using 1:300 molar dilution ratios. Falk and co-workers blended the whole mtDNA sequence and a standard 50 Mb whole exome, obtaining 99.8% of the mtDNA genome at 1000× coverage depth and a maximum sequencing depth of 10× for nuclear genomic regions. However, this nuclear coverage depth is not sufficient for diagnostic purposes, according to current guidelines [42]. Our data, in line with Abicht A. et al., demonstrate that by focusing the analysis on 257 nuclear genes, the coverage sequencing for nuclear genes reached the threshold for diagnostic purposes of 30× [33,43]. Furthermore, the nuclear gene panel analysis achieved the type B testing assessment, as required by the EJHG guidelines for diagnostic next-generation sequencing [42]. Moreover, for mtDNA sequences, we obtained a mean coverage >5000×. For NGS of mtDNA, an average coverage of 3000–4000× for WGS, with >90% of the genome covered to >2000×, is reported by Ryan L. Davis et al. [33]. L.J. Jennings and colleagues recommend a mean coverage of >5000 for the detection of somatic mutations to reliably detect heteroplasmic variants [43]. However, many factors influence the required depth of coverage, including the sequencing platform and the sequence complexity of the target, and the coverage for every NGS assay should be evaluated during assay development and validation.

Finally, one aspect of our study, which we believe could be interesting, is the preliminary assessment of the suitability of DNA from different tissue. The high-quality data obtained from these samples (tissue from slide and formalin-fixed paraffin-embedded tissue section samples) is an important achievement of the current NGS assay. We plan to validate these results in the upcoming experiments, including further samples from MD patients.

Following the validation phase, we applied the new analytical strategy to a child affected by severe early-onset optic atrophy. Early onset (6 years old) of severe optic atrophy led clinicians to suspect possible mitochondrial disease. We identified a mutation in the *OPA1* gene. The *OPA1* gene (OMIM 605290) encodes a protein that localizes to the inner mitochondrial membrane and regulates several cellular processes, including mitochondrial stability, mitochondrial bioenergetics output, and the sequestration of pro-apoptotic cytochrome c oxidase molecules within the mitochondrial cristae. Mutations in this gene are responsible for 60–80% of optic atrophy 1 cases, with autosomal dominant transmission characterized by moderate to severe loss of visual acuity. The age of onset varies from birth to more than 60 years [44,45,46]. Clinically unaffected parents have refused to submit to the genetic analysis; therefore, we could not investigate the segregation of the mutation in the family. However, the rs80356530 mutation is known to be associated with reduced penetrance, although in some cases, it is reported as a *de novo* mutation [47,48]. The molecular diagnosis of OPA1 deficit allowed us to start an appropriate therapy using a new pharmacological approach to improve mitochondrial energy production.

## 5. Study Limitations

The main limitation of our paper is related to the limit of heteroplasmy detection. In fact, mtDNA mutations may be heteroplasmic, with a coexistence of mutated and wild-type mtDNA inside an individual cell. The threshold at which the defect occurs depends on the specific mutation and on the cell type but is generally thought to be between 60% and 80% [2]. Typically, a high percentage of mutated mtDNA (>50%) is required to cause cellular defects; however, several mtDNA mutations (particularly mt-tRNA mutations) impact the phenotype only if present at very high levels [2,38]. Furthermore, the DNA source may be a critical variable in detecting heteroplasmy. In fact, blood DNA may have low heteroplasmy, unlike other affected tissues, not detectable by PCR and Sanger sequencing, which has a detection limit ranging between 10 and 20% [49,50]. Our data demonstrated that the custom 1:300 MITO-NUCLEAR design was able to detect heteroplasmic mutation of about 40%. In the analyzed samples, there was no variant with a level of heteroplasmy lower than 40%. However, the mtDNA genome was covered at least 3000×, reaching a mean coverage of >5000×. These analytic features suggest that we may be able to detect even very low levels of heteroplasmy, as suggested by the paper of Jenning et al. [43]. We should have verified the heteroplasmy detection limit by analyzing successive dilutions of the samples showing heteroplasmic variants. This will be a future update to our paper. Another limitation of the study is the small number of NGS experiments undertaken. We are currently already working on an implementation of our study, including samples from patients with suspected mitochondrial disease, to confirm the reproducibility and robustness of the MITO-NUCLEAR assay in order to demonstrate its diagnostic utility.

## 6. Conclusions

In conclusion, we demonstrated that the custom 1:300 Agilent SureSelect MITO-NUCLEAR Panel (SSel Custom Constitutional Panel 16.708 Kbp and nuclear clinic exome CCP17: Custom Constitutional Panel; Design size: 17 Mb) is an NGS-based platform for the parallel analysis of nuclear genes and the complete mtDNA and provides a potential one-step solution that can be applied to both research and clinical genetic diagnostic evaluations of patients with suspected mitochondrial disease, allowing the simultaneous discovery both of nuclear and mitochondrial mutations.

Furthermore, the clinical case reported in this paper highlights how NGS-based methods are revolutionizing the diagnosis of rare complex diseases, representing one of the most essential steps to enable a definitive diagnosis and to establish the etiology of diseases.

## Figures and Tables

**Figure 1 genes-14-01087-f001:**
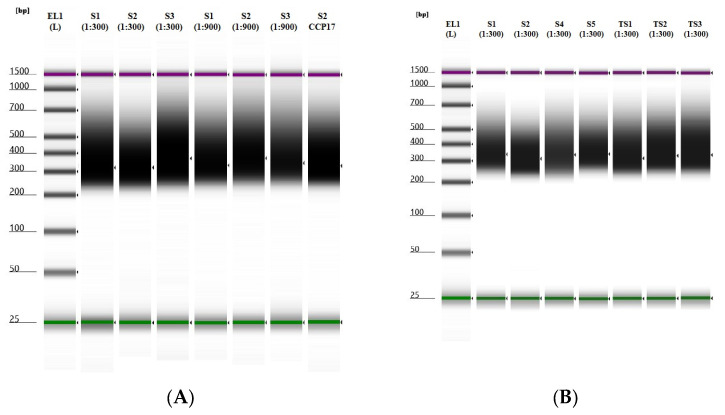
Post-capture quality profiles of NGS libraries were obtained using an Agilent TapeStation and High Sensitivity D1000 ScreenTape from two different molar dilution ratios, 1:300 and 1:900 (mitochondrial probes: nuclear probes). (**A**) RUN 1 experiments: S1, S2, and S3 samples were hybridized with a mix of both mitochondrial and nuclear probes at two different dilutions, 1:300 and 1:900 (mitochondrial probes: nuclear probes). Sample S2 was also hybridized with only nuclear probes (CCP17). (**B**) RUN 2 experiments: S1, S2, S4, S5, TSI, TS2, and TS3 samples were hybridized with a mix of both mitochondrial and nuclear probes at a 1:300 molar dilution ratio. EL1: Electronic Ladder.

**Figure 2 genes-14-01087-f002:**
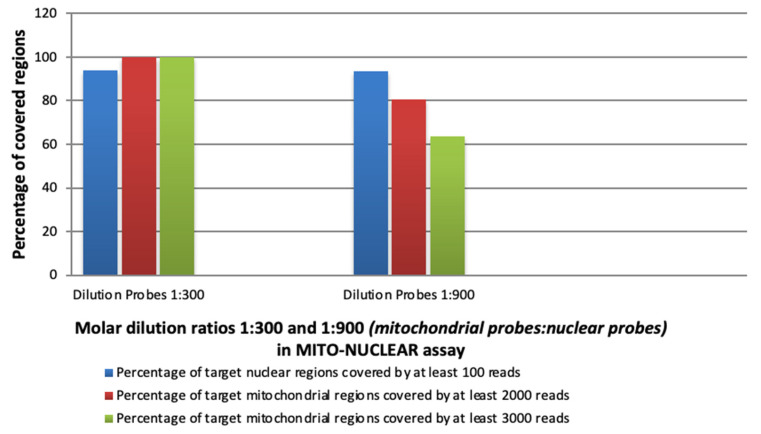
Capture efficiency of mtDNA (SSel Custom Constitutional Panel 16.708 Kbp) and nuclear clinic exome (CCP17: custom constitutional panel; Design size: 17 Mb for nuclear clinic exome) in MITO-NUCLEAR blended design into two different molar ratios of mtDNA probes. The total of 10 samples (S1, S2, and S3 from RUN 1 and S1, S2, S4, S5, TS1, TS2, and TS3 from RUN 2) are tested into the 1:300 molar ratios of mitochondrial probes. The three S1, S2, and S3 samples from RUN 1 are also tested into the 1:900 molar ratios of mitochondrial probes. The height of the columns indicates the percentage of nuclear regions (blue) and mtDNA regions (red/green) covered by a minimum number of reads.

**Figure 3 genes-14-01087-f003:**
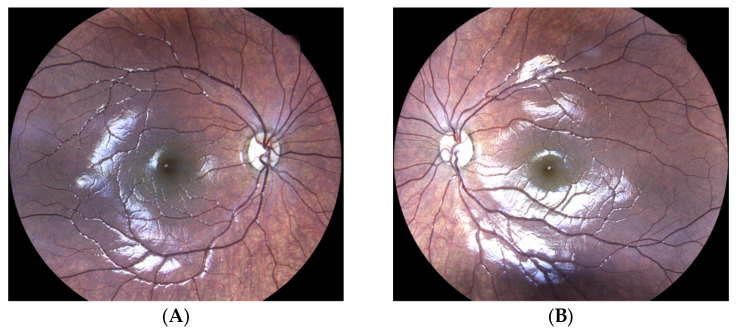
Eye fundus examination showing pallor of discs in the right eye (**A**) and the left eye (**B**) of the affected proband.

**Table 1 genes-14-01087-t001:** Coverage performances for the blended MITO-NUCLEAR experiments, with varying molar ratios (1:900 and 1:300) of custom probes for the mtDNA genome.

	ID	Tissue Type	Sample Type	Dilution mtDNA Probes	Bed Analysed	Average Reading Depth (Coverage)
**RUN 1**	S1	Blood	MITO-NUCLEAR	1.300	Nuclear CCP17	126
			MITO-NUCLEAR	1.300	mtDNA	5655
	S1	Blood	MITO-NUCLEAR	1.900	Nuclear CCP17	142
			MITO-NUCLEAR	1.900	mtDNA	3131
	S2	Blood	MITO-NUCLEAR	1.300	Nuclear CCP17	165
			MITO-NUCLEAR	1.300	mtDNA	3846
	S2	Blood	MITO-NUCLEAR	1.900	Nuclear CCP17	143
			MITO-NUCLEAR	1.900	mtDNA	1616
	S3	Blood	MITO-NUCLEAR	1.300	Nuclear CCP17	150
			MITO-NUCLEAR	1.300	mtDNA	4607
	S3	Blood	MITO-NUCLEAR	1.900	Nuclear CCP17	156
			MITO-NUCLEAR	1.900	mtDNA	2329
	S2	Blood	NUCLEAR CCP17	ND	Nuclear CCP17	155
**RUN 2**	S1	Blood	MITO-NUCLEAR	1.300	Nuclear CCP17	140
			MITO-NUCLEAR	1.300	mtDNA	7000
	S2	Blood	MITO-NUCLEAR	1.300	Nuclear CCP17	158
			MITO-NUCLEAR	1.300	mtDNA	3920
	S4	Swab	MITO-NUCLEAR	1.300	Nuclear CCP17	162
			MITO-NUCLEAR	1.300	mtDNA	3700
	S5	Swab	MITO-NUCLEAR	1.300	Nuclear CCP17	149
			MITO-NUCLEAR	1.300	mtDNA	3280
	TS1	Fresh Tissue	MITO-NUCLEAR	1.300	Nuclear CCP17	155
			MITO-NUCLEAR	1.300	mtDNA	7000
	TS2	Tissue from slide	MITO-NUCLEAR	1.300	Nuclear CCP17	163
			MITO-NUCLEAR	1.300	mtDNA	5127
	TS3	FFPE	MITO-NUCLEAR	1.300	Nuclear CCP17	125
			MITO-NUCLEAR	1.300	mtDNA	11,000

MITO-NUCLEAR, MITOchondrial genome (ChrMito hg19 Design size: 16.708 Kbp), and NUCLEAR clinic exome (CCP17 Design size: 17.692 Mb) were blended into two different molar ratios of mitochondrial probes (1:900 and 1:300) (RUN 1) and a molar ratio of mitochondrial probes 1:300 (RUN 2). Sample S2 was also tested only with CCP17 design; ND, no dilution; mtDNA, mitochondrial genome; CCP17, custom constitutional panel; Design size: 17 Mb; FFPE, formalin-fixed paraffin-embedded.

## Data Availability

All research data related to molecular analysis and clinical management of patients reported in the paper can be requested from the corresponding authors.

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
