# Peer review of "Combined MITOchondrial-NUCLEAR (MITO-NUCLEAR) Analysis for Mitochondrial Diseases Diagnosis: Validation and Implementation of a One-Step NGS Method"

_genes, 2023, doi:10.3390/genes14051087_

Round 1
Reviewer 1 Report
The authors describe preliminary validation of a combined Next Generation Sequencing approach to analyse genes encoded by mitochondrial DNA and nuclear DNA in a single assay. Although not novel, there are surprisingly few publications in the literature utilising this approach and so this study has potential value. The method adopted builds on work reported by Falk et al. 2012.
Unfortunately, the study is limited by the small amount of data (only 2 small NGS runs were undertaken) and also includes a number of errors. In my view, a significant amount of additional data is probably required. For example, the study would benefit from a full validation (e.g. confirmation of detection of >=300 different sequence variants in both mtDNA and nuclear DNA to provide high confidence of test sensitivity) including assessment of limit of heteroplasmy detection, and/or diagnostic genetic analysis of a mitochondrial disease cohort. The study includes a preliminary assessment of the suitability of DNA from different tissue types and it is very encouraging that high quality data was obtained from these samples; the authors may wish to expand on this in future work and could have discussed this further.
It is also important to note that Whole Genome Sequencing is increasingly being adopted for genetic/genomic diagnostic testing around the world, and allows simultaneous analysis of mtDNA and nuclear DNA.
In addition, I have provided the following detailed comments on the manuscript:
· Lines 44-45 of the introduction: This sentence requires rewording as currently it implies that all MDs are due to mtDNA mutations.
· Lines 48-50 of the introduction: cardiomyopathies and cardiac conduction defects are largely separate, and so cardiac conduction defects should not be referred to as an example of cardiomyopathy.
· Line 80 of the introduction: remove the word “until”.
· Lines 112-115 of Materials and Methods: Samples S2, S2 & S4 should not be described as having pathogenic mtDNA variants (lines 114-115), as, in the results (lines 240-244), these variants are stated to be m.4126T>C and m.4917A>G. These two variants are not generally accepted as causative of primary mitochondrial disease and would be classified as benign variants by most mitochondrial disease diagnostic centres (due to their high population frequency). Therefore, these samples should not be referred to as “from patients with MD molecular diagnosis” (line 112) although they are still valid samples to compare sequencing results from this study to those from previous sequencing analyses.
· Line 113 of Materials and Methods: “mitochondrial variants” should be “mitochondrial DNA variants” or “mtDNA variants”.
· Lines 122-124 of Materials and Methods: I can’t see any information here or elsewhere in the manuscript regarding the tissue type for samples TS1, TS2 and TS3; is this skeletal muscle tissue for example?
· Line 199 of Materials and Methods: I think by “familiarity” the authors mean “family history”.
· Line 213 of Results (and also lines 31-32 of the abstract): “100% of the mtDNA genome was covered at least 3,000X” appears to be a misleading statement as line 214 states that the “minimum depth of coverage for any mtDNA base was 140X”. These statements appear contradictory to me, so clarification is required: if some bases were only covered at 140X, how can the entire mtDNA be covered at 3,000X?
· Lines 238-245 of Results: A summary of the total number of mtDNA variants (including benign polymorphisms) correlated with previous results would be helpful, which could be supported by supplementary data. Also, were all variants homoplasmic apart from m.3243A>G in S5?
· Line 240 of Results: “omoplasmic” should be “homoplasmic”.
· Lines 241-246: HNGC gene names should be used and italicised, for example “MT-ND1” (in italics); also “MT-TL1” for tRNALeu(UUA/G).
· Lines 258-262 of Results: It would be helpful to provide the read split/count data for the m.3243A>G heteroplasmic variant call. Also, was this the only heteroplasmic variant detected across all samples?
· Line 253 of Results: Was the SUCLG1 variant homozygous or heterozygous? Also, a reference transcript should be specified. The authors should make it a clear that this is a variant of uncertain significance or likely benign variant and so does not indicate a diagnosis of SUCLG1-related mtDNA depletion syndrome.
· Lines 283-286 of Results: The text should not be in italics.
· Line 288 of Results: A reference transcript should be specified for OPA1.
· Lines 297-298 of Discussion: Given the low limit of heteroplasmy detection associated with Sanger sequencing and the advent of NGS, it is not true to say that “the most reliable and sensitive method for detecting mitochondrial DNA variants is long-range PCR, followed by Sanger sequencing”.
· Lines 310-312 of Discussion: This statement is not true for all mitochondrial diagnostic centres internationally; the authors have already referred to the work of Falk et al. 2012 and Abicht et al. 2018. Also, whole genome sequencing approaches can be used to simultaneously sequence both mitochondrial and nuclear genomes, e.g. Davis et al. 2022 (PMID: 35641312).
· Lines 333-335 and 338-349 of Discussion: More detail is required and I cannot find mention of required mean coverage of 5,000X in the references provided. Minimum read coverage is more relevant to sensitivity than mean coverage and the relationship between these two depends on uniformity of coverage across mtDNA. The authors should comment on the limit of heteroplasmy detection that they believe their assay achieves. The second paragraph (lines 338-349) should include some explanation as to why “some MDs may result from heteroplasmy levels not detectable by PCR and Sanger sequencing”, i.e. due to the source of DNA available for testing, typically blood DNA which may have low heteroplasmy when heteroplasmy is much higher in affected tissues.
· Line 335 of Discussion: “achived” should be “achieved”.
· Line 352 of Discussion: Given that the OPA1 variant detected is a relatively common pathogenic OPA1 variant present at a low frequency in the general population and known to be associated with reduced penetrance, it is not appropriate to describe it as “possible de novo”; this could be commented on as an unlikely possibility, but it should be made clear that it is most likely to have been inherited from one of the parents.
· Lines 374-376 of the Conclusion: The OPA1 case is not a powerful example of the utility of this NGS panel, as historic genetic testing for optic atrophy would also have readily identified this variant.
· The entire manuscript should also be thoroughly reviewed for typographical and grammatical errors.
Reviewer 2 Report
The authors presented an article describing the design and validation of a custom NGS panel, which allows for simultaneous detection of genetic variants in both mtDNA and mitochondrial-related nuclear genes. The NGS panel was successfully used to arrive at a genetic diagnosis in a young patient presenting with severe early-onset retinopathy due to a heterozygous mutation on the nuclear gene OPA1.
Comments: The originality of the manuscript, giving a detailed description of the design, evaluation and validation process of a custom panel for the simultaneous detection of variants in nuclear genes and mtDNA at adequate depth in both genomes makes the article, from my perspective, worth publishing. However, some minor points should be addressed regarding the validation process of the custom NGS panel:
- During the validation process, only one heteroplasmic mutation in mtDNA was evaluated in one tissue. Although high-depth NGS sequencing is assumed to be a reliable method for measuring heteroplasmy in mtDNA mutations, it would be advisable to validate a greater number of mtDNA mutations in different tissues and at different heteroplasmic levels (low, medium, and high) to ensure reproducibility in the accuracy of the panel's measurement of heteroplasmy.
- The detection limit for measuring heteroplasmy of mutations in mtDNA is a relevant topic due to, for example, the presence of low levels of heteroplasmy in some high-turnover tissues such as blood, but which may be present at a high percentage in other tissues such as muscle and brain. In this regard, it would be interesting to know the detection limit of the MITO-NUCLEAR panel and how this detection limit has been calculated (taking into account the depth of coverage achieved, bioinformatics pipeline applied, and sequencing error rate).
Round 2
Reviewer 1 Report
This paper reports preliminary validation of a combined Next Generation Sequencing approach to analyse genes encoded by mitochondrial DNA and nuclear DNA in a single assay. In this resubmission, the authors have addressed most of the points previously raised. Although they have not undertaken additional studies to extend the work to a full validation that would include determination of limit of mtDNA heteroplasmy detection, the authors have now clearly discussed the limitations of their study and included more detail of their existing validation data (in Supplementary Table 1).
In my view, the following minor issues, which largely relate to English language, still require addressing:
Line 47: Suggest changing “several nuclear gene mutations are also responsible of” with “numerous nuclear gene mutations are also associated with”.
Line 58: “than thousand” should be “than a thousand”.
Line 83: In view of comments that some dual genome assays are in use, suggest amending to “investigation by NGS typically still requires analysing”.
Line 103: “validation’s experiments” should be simply “validation experiments”.
Line 138: I think the word “approval” is missing after “committee”.
Line 293: “none known mutations” should be “no known mutations”.
Line 301: Please refer to Supplementary Table 1 to support the statement that all nuclear variants were confirmed in sample S3.
Line 307: “Read Deep” should be “Read Depth”.
Line 312: “previously” should be “previous”.
Line 340: Remove “The” before “98%”.
Line 341: “reading” should be “read”.
Line 354: As diagnosis of mtDNA defects does not necessarily “need” the use of multiple methods, please amend, for example, to “may utilise”.
Line 368: Suggest replacing “such as the simultaneous analyses of group of nuclear genes” with “as well as the simultaneous analysis of a group of nuclear genes”.
Lines 375-376: This sentence needs rewording along the following lines “Moreover, whole genome sequencing approaches have recently been used to simultaneously sequencing both mitochondrial and nuclear genomes.”
Line 380: “Mitomaps” should be “Mitomap”.
Lines 433 and 434: I don’t think this child’s disorder should be described as a “retinopathy”; please amend to “visual loss” or “optic atrophy”.
Line 466: Suggest replacing “portend” with “suggest” or “indicate”.
Lines 452 to 470: The authors should also comment that another limitation of their study is the small number of NGS runs (only two) undertaken and so future work will include carrying out further assays to confirm reproducibility and robustness; also in future work, the authors should ideally test more samples from patients with suspected mitochondrial disease in order to further demonstrate diagnostic utility.
